# Beyond Expert-Annotated Labels: An Adaptive Label Learning Method for Knowledge Tracing

## Abstract

Knowledge Tracing (KT) serves as an indispensable technology in intelligent tutoring systems (ITS), aiming to predict learners' future performance based on their past interactions. Current KT models commonly use predefined knowledge concept (KC) labels to improve prediction accuracy. These labels provide grouping information about questions, allowing models to infer learners' performance on low-frequency questions. However, the subjectivity of human labeling may not accurately reflect which questions share similar cognitive processes, potentially limiting the models' performance. To address this, we redefine KT as a problem of learning from question groupings and introduce an adaptive framework that iteratively refines groupings through alternating optimization. We initiate with random groupings and freeze them to optimize the KT model with gradient descent, then select the loss-minimizing configuration by computing the loss for each possible reassignment of questions to different groups under continuous assignment probabilities, repeating this process until convergence. We evaluate our approach on real-world ITS datasets, incorporating the optimized groupings into different KT models instead of KCs, which markedly improves model performance and achieves state-of-the-art results. Further experiments uncover the underlying semantic connections between our automatic groupings and prior KCs, revealing potential similarities in cognitive mechanisms among KCs, providing new insights and research directions for educational and cognitive sciences. **Code is available in the supplementary materials**.

## 1 Introduction

Knowledge Tracing (KT), as a core component of intelligent tutoring systems (ITS) (Lin et al., 2023), is essentially a sequential prediction task. It aims to predict the future performance of learners based on their past interaction records, which involves determining whether they will answer a given question correctly. By analyzing historical data to assess learners' current knowledge states and identify areas needing reinforcement, KT models support the ITS in providing targeted feedback and adapting the learning content to the needs and preferences of each learner (Huang et al., 2019; Liu et al., 2019), thereby promoting their progress and maximizing their potential for achievement (Abdelrahman et al., 2023).

Early KT models (Corbett & Anderson, 1994; Cen et al., 2006; Thai-Nghe et al., 2012) typically rely on predefined rules and fixed parameters, which fail to accurately model the complex interactions of learners, often resulting in poor fit and weak predictive power. In recent years, the integration of deep learning techniques such as LSTM (Piech et al., 2015; Yeung & Yeung, 2018), Transformers (Pandey & Karypis, 2019; Ghosh et al., 2020), and GNNs (Nakagawa et al., 2019; Yang et al., 2020; Cheng et al., 2024) has significantly enhanced the pattern recognition capabilities of KT models, enabling them to better adapt to long learning sequences. This enhancement improves precision and reliability, bringing KT closer to realizing its maximum potential. Subsequently, researchers have turned to educational-psychology theories, introducing assumptions about phenomena such as learning behavior (Xu et al., 2023), memory (Zhang et al., 2017; Li et al., 2023) and forgetting (Nagatani et al., 2019; Abdelrahman & Wang, 2022) into KT models, which leads to further improvements in accuracy. All relevant studies reveal that the success of these models depends on high-quality

data, especially the importance of question labels such as knowledge concepts (KCs). These KCs are manually assigned and provide the models with reliable prior knowledge about which questions are closely related.

Educational measurement theory (Desmarais et al., 2012; Shi et al., 2023) emphasizes that the relationship between questions is intricate, as a single question may draw on multiple KCs and the skills required to solve one question can often be applied to others. Labeling questions essentially groups them, a process that involves considering a wide array of combinations and makes finding the optimal arrangement challenging. Moreover, this process is often based on the intuition of domain experts rather than data-driven patterns, which can introduce subjective biases and may not accurately reflect the cognitive processes exhibited by learners in actual problem-solving situations. Consequently, the noise and inaccuracies in manually assigned labels can hinder downstream models' ability to learn accurate patterns, leading to newly proposed models introducing additional parameters and adopting more complex structures.

Exploring the optimal grouping of questions is a combinatorial optimization problem with a vast solution space. This problem is closely related to Stirling numbers (Rennie & Dobson, 1969), which enumerate the ways to partition a set into non-empty subsets, thereby underscoring the complexity of identifying an optimal arrangement. Given this, we believe it is unlikely that manual KC labels would just happen to be the optimal grouping for questions, and it is highly likely that superior groupings exist. To avoid the impracticality of exhaustive searching, a more practical approach is to develop a data-driven and KT task-oriented adaptive grouping algorithm for questions. This algorithm can dynamically adjust the grouping of questions to maximize the performance of KT models on specific datasets.

Specifically, we propose an adaptive label learning method for knowledge tracing (ALL4KT), which explores optimal question grouping through alternating minimization. We begin with a random question assignment matrix and iteratively update the model parameters using gradient descent while keeping the matrix fixed. Next, we refine the question grouping by simulating reassignments to different groups and selecting the one that yields the lowest loss. To avoid greedy assignments, we employ a relaxation strategy to transform the problem into a continuous probability space, thereby balancing exploration and exploitation. This iterative process continues until the assignment matrix stabilizes. Ultimately, we incorporate the grouping results into various baseline KT models and evaluate them on four real ITS datasets. The results demonstrate that our method significantly boosts the performance of each of the KT models and reveals the underlying relationships among KCs through group semantic analysis.

## 2 RELATED WORK

Most KT research is conducted on the basis of KCs (Lu et al., 2022; Zhang et al., 2025), which effectively reflect the characteristics of real-world educational content (Zanellati et al., 2024). Piech et al. (2015) introduced DKT, pioneering the use of deep neural networks in KT by directly employing KCs instead of questions. By leveraging the strengths of deep learning, DKT achieved significant improvements over early methods that relied on predefined rules and fixed parameters (Corbett & Anderson, 1994; Cen et al., 2006; Thai-Nghe et al., 2012). DKVMN (Zhang et al., 2017) maps KC labels to static keys and dynamic values using a dynamic key-value memory network, effectively modeling learners' mastery of knowledge concepts. Subsequent research (Zhou et al., 2025; Wang et al., 2025) has advanced the utilization of KC labels by integrating various educational psychology theories, enhancing model performance. For instance, AKT (Ghosh et al., 2020) combines the Rasch model with attention mechanisms to capture relationships between questions and strengthen the modeling of time-sensitive learning behaviors. ReKT (Shen et al., 2024), inspired by the decision-making processes of human teachers, proposes a lightweight forget-response-update framework that utilizes KC labels to guide knowledge state updates. UKT (Cheng et al., 2025) introduces uncertainty modeling, FlucKT (Hou et al., 2025) focuses on short-term cognitive fluctuations, and LefoKT (Bai et al., 2025) incorporates relative forgetting attention. These methods deepen the utilization of KC labels from different perspectives, driving KT research towards greater precision and interpretability.

Although modeling focused on KCs has become a consensus in KT, the reliability of KC labels often depends on expert annotation, which can introduce errors and inconsistencies. Some questions

involve multiple overlapping or implicit KCs, and overly fine-grained labels may lead to data sparsity, while overly broad labels may not accurately capture nuanced cognitive demands. These issues highlight the limitations of relying solely on KC labels for KT and emphasize the need for more robust and flexible methods to complement their use.

# 3 PROPOSED METHOD

## 3.1 PROBLEM FORMULATION

The ITS encompasses several entities (Liu et al., 2023a), including a set of learners represented by $\mathcal{U}$ and a set of questions represented by $\mathcal{Q}$. For any learner $u \in \mathcal{U}$, their answer sequence is represented as $\mathcal{S}_u \triangleq \{(q_i^{(u)}, r_i^{(u)})\}_{i=1}^{T_u}$, where $q_i^{(u)} \in \mathcal{Q}$ denotes the question encountered by learner $u$ during the $i$-th attempt. $r_i^{(u)} \in \{0, 1\}$ indicates the learner's response to the question, where 0 means incorrect and 1 means correct. $T_u \in \mathbb{Z}^+$ represents the number of attempts made by the learner, which is also the total number of time steps. The historical records of learner $u$ prior to the $t$-th attempt can be denoted as $\mathcal{S}_{u,<t}$. The probability of learner $u$ answering the $t$-th question correctly is expressed as $\hat{r}_t^{(u)} \triangleq \mathbb{P}(r_t^{(u)} = 1 \mid \mathcal{S}_{u,<t}, q_t^{(u)})$. Given a dataset $\mathcal{D} \subseteq \{(u, q, r, t) \mid u \in \mathcal{U}, q \in \mathcal{Q}, r \in \{0, 1\}, t \in \mathbb{Z}^+\}$, a flattened representation of all historical data, we have

$$\mathcal{D} = \bigcup_{u \in \mathcal{U}} \bigcup_{t=1}^{T_u} \left\{(u, q_t^{(u)}, r_t^{(u)}, t)\right\}.$$

Questions, often grouped based on KCs, endow the model with richer semantic information, particularly when data are sparse. We denote the set of question groups as $\mathcal{G}$, and introduce the question assignment matrix $\mathbf{Z} \in \{0, 1\}^{|\mathcal{Q}| \times |\mathcal{G}|}$, where $\mathbf{Z}_{q,g} = 1$ indicates that question $q$ is assigned to group $g$. For all $q \in \mathcal{Q}$, it holds that $\sum_{g \in \mathcal{G}} \mathbf{Z}_{q,g} = 1$. Given a KT model with all trainable parameters denoted by $\boldsymbol{\theta} \in \Theta$, The expression for the probability that learner $u$ answers question $q$ correctly at time $t$ is

$$p_{u,q,t}(\boldsymbol{\theta}, \mathbf{Z}) \triangleq \mathbb{P}(r = 1 \mid \mathcal{S}_{u,<t}, q, \boldsymbol{\theta}, \mathbf{Z}).$$

Thus, we can intuitively define the cross-entropy loss function for a single sample as

$$\ell_{u,q,t}(\boldsymbol{\theta}, \mathbf{Z}) = -r \ln p_{u,q,t}(\boldsymbol{\theta}, \mathbf{Z}) - (1 - r) \ln(1 - p_{u,q,t}(\boldsymbol{\theta}, \mathbf{Z})).$$

Our ultimate goal is to minimize the overall cross-entropy loss by optimizing the question assignment matrix $\mathbf{Z}$ and the model parameters $\boldsymbol{\theta}$, which can be formulated as

$$\min_{\mathbf{Z}, \boldsymbol{\theta}} \quad \mathcal{L}(\boldsymbol{\theta}, \mathbf{Z}) = \sum_{(u,q,r,t) \in \mathcal{D}} \ell_{u,q,t}(\boldsymbol{\theta}, \mathbf{Z}).$$

Obviously, model parameters $\boldsymbol{\theta}$ are continuous and differentiable, while the matrix $\mathbf{Z}$ comprises discrete variables, which requires searching for the optimal solution in a discrete space. For the former, the gradient descent method can be employed, while for the latter, it is an NP-complete problem. Finding the optimal $\mathbf{Z}$ is particularly challenging, as this problem often has multiple local optima. A common strategy is to use alternating optimization (Bezdek & Hathaway, 2003), iteratively refining the solution to get closer to the optimum.

## 3.2 ALTERNATING MINIMIZATION

Alternating minimization is a common algorithm for multivariable optimization problems that works by iteratively fixing some variables and optimizing the remaining ones to progressively explore the optimal solution. The iterative process in our problem can be summarized as fixing the question assignment matrix $\mathbf{Z}$ to update the parameters $\boldsymbol{\theta}$ of the KT model and then fixing the parameters $\boldsymbol{\theta}$ to update $\mathbf{Z}$. The initialization of $\mathbf{Z}$ involves randomly assigning each question $q \in \mathcal{Q}$ to a group

label $g_q$ from the set of group indices $\{1, 2, \ldots, |\mathcal{G}|\}$, with the initial assignment matrix $\mathbf{Z}^{(0)}$ defined as follows:

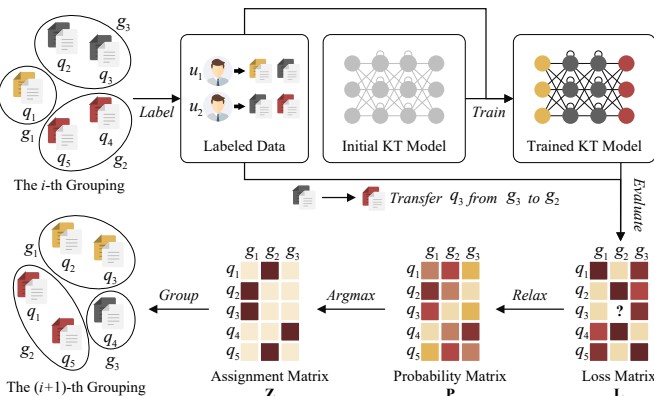

The $i$-th Grouping

Label — Labeled Data — Initial KT Model — Train — Trained KT Model

Transfer $q_3$ from $g_3$ to $g_2$

The $(i+1)$-th Grouping — Group — Assignment Matrix $\mathbf{Z}$ — Argmax — Probability Matrix $\mathbf{P}$ — Relax — Loss Matrix $\mathbf{L}$ — Evaluate

$$\mathbf{Z}_{q,g}^{(0)} = \begin{cases} 1, & g = g_q, \\ 0, & \text{otherwise.} \end{cases}$$

Subsequently, during the iterative process, the current question assignment matrix $\mathbf{Z}^{(i)}$ is treated as a constant, reducing the original problem to a conventional KT training task involving only the optimizable continuous variables $\boldsymbol{\theta}$:

Figure 1: We utilize the $i$-th grouping information to relabel questions in the original dataset, replacing KCs to train a KT model until convergence, and then reassign each question to different groups to compute the loss matrix via the KT model. Subsequently, we apply relaxation operation to the loss matrix to derive a probability matrix, and finally select the highest-probability group for each question as the $(i + 1)$-th round of grouping.

$$\boldsymbol{\theta}^{(i+1)} = \arg\min_{\boldsymbol{\theta} \in \Theta} \mathcal{L}(\boldsymbol{\theta}, \mathbf{Z}^{(i)}).$$

The problem can be solved using any KT model, such as DKT, in conjunction with any gradient-based optimization algorithm, like Adam (Kingma & Ba, 2015). In practice, we sample learners in batches and perform forward and backward propagation to update the parameters, thereby obtaining the updated $\boldsymbol{\theta}^{(i+1)}$. Then, by fixing $\boldsymbol{\theta}^{(i+1)}$, we compute the potential loss for each question $q$ when assigned to any group $g \in \mathcal{G}$, as follows:

$$\mathcal{L}_{q,g}^{(i+1)} = \sum_{(u,q,r,t) \in \mathcal{D}} \ell_{u,q,t}\big(\boldsymbol{\theta}^{(i+1)}, \mathbf{Z}^{(i)}[\mathbf{Z}_{q,g}^{(i)} \leftarrow 1]\big),$$

We can assess the impact of different combinations on the global loss by locally substituting the association between question $q$ and group $g$ in the matrix $\mathbf{Z}^{(\mathbf{i})}$. $\mathcal{L}_{q,g}^{(i+1)}$ is regarded as the cost of placing question $q$ into group $g$, and the loss matrix $\mathbf{L}^{i+1}$ is constructed by calculating the corresponding losses for all $q \in \mathcal{Q}$ and $g \in \mathcal{G}$. Ultimately, the allocation is obtained through argmin hardening:

$$\mathbf{Z}_{q,g}^{(i+1)} = \mathbb{I}\left\{ g = \arg\min_{g'} \mathcal{L}_{q,g'}^{(i+1)} \right\},$$

where $\mathbb{I}\{\cdot\}$ is the indicator function, taking the value 1 if the condition is met and 0 otherwise. As analyzed in Appendix A.1, by fixing $\mathbf{Z}^{(i)}$, the model is trained to ensure that there exists at least one set of $\boldsymbol{\theta}^{(i+1)}$ such that $\mathcal{L}(\boldsymbol{\theta}^{(i+1)}, \mathbf{Z}^{(i)}) \leq \mathcal{L}(\boldsymbol{\theta}^{(i)}, \mathbf{Z}^{(i)})$. Subsequently, by fixing $\boldsymbol{\theta}^{(i+1)}$, we can obtain a set of $\mathbf{Z}^{(i+1)}$ such that $\mathcal{L}(\boldsymbol{\theta}^{(i+1)}, \mathbf{Z}^{(i+1)}) \leq \mathcal{L}(\boldsymbol{\theta}^{(i+1)}, \mathbf{Z}^{(i)})$. Therefore, the sequence $\{\mathcal{L}(\boldsymbol{\theta}^{(i)}, \mathbf{Z}^{(i)})\}_{i=0}^{\infty}$ theoretically satisfies

$$\mathcal{L}(\boldsymbol{\theta}^{(i+1)}, \mathbf{Z}^{(i+1)}) \leq \mathcal{L}(\boldsymbol{\theta}^{(i)}, \mathbf{Z}^{(i)}),$$

with the loss function having a lower bound of 0. Consequently, the sequence converges to a limit $(\mathbf{Z}^*, \boldsymbol{\theta}^*)$ such that $\mathcal{L}(\boldsymbol{\theta}^*, \mathbf{Z}^*) \leq \mathcal{L}(\boldsymbol{\theta}, \mathbf{Z}^*)$, $\mathcal{L}(\boldsymbol{\theta}^*, \mathbf{Z}^*) \leq \mathcal{L}(\boldsymbol{\theta}^*, \mathbf{Z})$. In practice, the algorithm terminates and outputs $(\mathbf{Z}^*, \boldsymbol{\theta}^*)$ if $\mathbf{Z}^{(i+1)} = \mathbf{Z}^{(i)}$ or the iteration count $i \geq I_{\max}$, and while it does not guarantee finding the global optimum, it offers a practical and efficient approach to iteratively approximate the optimal solution.

### 3.3 ASSIGNMENT RELAXATION

Although alternating minimization can theoretically iteratively approximate the optimal solution, the hard assignment based on the loss matrix $\mathbf{L}$ is essentially a greedy strategy that may quickly converge to local optima or cause oscillations during the iterative process (Barik et al., 2025), resulting in an imbalance in the grouping, where some groups become too large while others remain too small. Inspired by the optimal transport problem (Cuturi, 2013; Pham et al., 2020), we propose a soft assignment strategy based on probabilistic representation, where questions are treated as supply points and groups as demand points, with questions needing to be assigned to groups. This approach balances the distribution of questions across different groups while ensuring that the overall assignment adheres to the constraints of supply and demand. Specifically, we transform the loss matrix $\mathbf{L}$ into a non-negative probability matrix $\mathbf{P}$, where each element $\mathbf{P}_{q,g}$ represents the probability of assigning question $q$ to group $g$. Our goal is to find an scheme that minimizes the total allocation cost, specifically to obtain $\mathbf{P}^\star$ such that

$$\mathbf{P}^\star = \arg\min_{\mathbf{P}\in\Delta}\langle\mathbf{P},\mathbf{L}\rangle + \lambda \cdot \mathrm{KL}(\mathbf{P}\|\mathbf{R}),$$

where $\langle\cdot,\cdot\rangle$ denotes the matrix inner product, reflecting the total cost of the assignment scheme. $\mathrm{KL}(\mathbf{P}\|\mathbf{R})$ is the Kullback-Leibler divergence, measuring the difference between $\mathbf{P}$ and the prior distribution $\mathbf{R}$, with the purpose of making the final assignment as close to the prior as possible. Here, each row $\mathbf{R}_q$ of the prior matrix $\mathbf{R}\in\Delta$ is defined as the initial probability distribution for question $q$. Setting $\mathbf{R}_{q,g} = 1/|\mathcal{G}|$ expresses a uniform, unbiased assumption, meaning each question has an equal chance of being assigned to any group. The temperature coefficient $\lambda$ determines the balance between trusting the cost and trusting the prior, where a larger $\lambda$ increases the KL term's penalty, making the probability assignment $\mathbf{P}^\star$ closer to a uniform distribution for smoother, more exploratory assignments, while a smaller $\lambda$ makes the cost term dominant, causing $\mathbf{P}^\star$ to become sharper and approach hard assignments. As derived in Appendix A.2, the closed-form solution can be written as

$$\mathbf{P}^\star = \mathrm{diag}(\mathbf{u})\,\mathbf{K}\,\mathrm{diag}(\mathbf{v}), \qquad \mathbf{K}_{q,g} = e^{-\mathbf{L}_{q,g}/\lambda}\,\mathbf{R}_{q,g},$$

where $\mathbf{u}$ and $\mathbf{v}$ are obtained through Sinkhorn row and column normalization iterations to ensure that the sum of each row and column is 1. We initialize $\mathbf{u} = \mathbf{1}_{|\mathcal{Q}|}$ and $\mathbf{v} = \mathbf{1}_{|\mathcal{G}|}$. Then, we iteratively update $\mathbf{u} \leftarrow \mathbf{1}_{|\mathcal{Q}|}/(\mathbf{K}\cdot\mathbf{v})$, $\mathbf{v} \leftarrow \mathbf{1}_{|\mathcal{G}|}/(\mathbf{K}^\top\cdot\mathbf{u})$, until convergence. Substituting these expressions into the formula yields $\mathbf{P}^\star$, and the allocation rule is given as follows:

$$\mathbf{Z}_{q,g}^{(i+1)} = \mathbb{I}\left\{g = \arg\max_{g'}\mathbf{P}_{q,g'}^{(i+1)}\right\}.$$

### 3.4 ALGORITHMIC COMPLEXITY

To comprehensively evaluate the feasibility of the proposed method in practical applications, we analyze the time complexity of each component in the alternating minimization process and conclude with an overall complexity assessment. Let $T_{\mathrm{avg}}$ denote the average length of the answer sequences across all learners, $D$ the dimension of the KT hidden state vector, $E$ the number of epochs for KT model training, $S$ the number of Sinkhorn iterations, and $I_{\max}$ the maximum number of iterations for alternating minimization. According to the analysis in Appendix A.3, after executing $I_{\max}$ iterations, the overall complexity is given by

$$\mathcal{O}\left(I_{\max}\cdot\left[\underbrace{E\cdot|\mathcal{U}|\cdot T_{\mathrm{avg}}\cdot D^2}_{\text{Model Training}} + \underbrace{|\mathcal{G}|\cdot|\mathcal{U}|\cdot T_{\mathrm{avg}}\cdot D^2}_{\text{Loss Matrix Computation}} + \underbrace{S\cdot|\mathcal{Q}|\cdot|\mathcal{G}|}_{\text{Sinkhorn Optimization}}\right]\right).$$

The product $|\mathcal{U}| \cdot T$, representing the number of interaction records in the data, is typically large, making model training and loss matrix computation the most significant factors in the overall process. When the number of groups $|\mathcal{G}|$ is small, the primary performance concern is training the KT

model; however, with a large $|\mathcal{G}|$, the time needed for grouping questions can significantly hinder performance. The complexities of Sinkhorn optimization and hard assignment are relatively low and typically do not pose a performance bottleneck.

# 4 EXPERIMENTS

## 4.1 EXPERIMENTAL SETTING

**Datasets**. We evaluate our model on four benchmark datasets: *ASSIST2009*[1] and *ASSIST2012*[2], collected from the ASSISTments platform (Feng et al., 2009) during the 2009–2010 and 2012–2013 school years, respectively, and *Algebra2005* and *Bridge2006*, originating from the KDD Cup 2010 EDM Challenge [3] (Stamper et al., 2010). Each dataset contains fields for learners, questions, KCs, and interactions, with statistics shown in the following table. In many datasets, a question may correspond to

Table 1: Statistics of all datasets.

| Datasets | Learners | Questions | KCs/Groups | Interactions |
|---|---|---|---|---|
| ASSIST2009 | 4,029 | 16,888 | 110/137 | 325,515 |
| ASSIST2012 | 28,118 | 53,084 | 265/265 | 2,710,913 |
| Algebra2005 | 574 | 17,2994 | 113/436 | 606,401 |
| Bridge2006 | 1,146 | 129,255 | 494/564 | 1,817,018 |

multiple KCs, and the number of these default groups is also detailed in the table (Note that the groups here are the default grouping caused by the KCs, not the groups set by our method).

**Baselines**. We compare our model against several state-of-the-art baselines, including *DKT* (Piech et al., 2015), *DKVMN* (Zhang et al., 2017), *AKT* (Ghosh et al., 2020), *SimpleKT* (Liu et al., 2023b), *ReKT* (Shen et al., 2024), *UKT* (Cheng et al., 2025), *FlucKT* (Hou et al., 2025), and *LefoKT* (Bai et al., 2025) (the codes for these baselines are sourced from Liu et al. (2022)). For detailed information on these baselines, see Appendix A.4.

**Hyperparameter Settings**. We conduct extensive grid search experiments, using powers of 2 as the metric, to adapt our hyperparameters to different datasets. We set the model dimension $D$ to 128. For the number of clusters, we assign 16 clusters for ASSIST2009 and Algebra2005, 128 clusters for ASSIST2012, and 512 clusters for Bridge2006. The $\lambda$ parameter is tuned to 4 for ASSIST2009, 16 for ASSIST2012, 128 for Algebra2005, and 32 for Bridge2006. The number of Sinkhorn iterations $S$ is set to 10, while the maximum number of iterations $I_{\max}$ in alternating minimization is set to 30. The learning rate is configured as 1e-3, and the batch size is set to 40.

**Evaluation Metrics**. We conduct experiments using five-fold cross-validation to calculate the Area Under the Curve (AUC), Accuracy (ACC), and Root Mean Squared Error (RMSE) for each model on each dataset. The probability matrix $\mathbf{P}$ is computed using only the training set data, and the model performance is ultimately evaluated on the test data.

**Experimental Environment**. Our experiments are carried out on a server equipped with an Intel Xeon Platinum 8358P CPU, 128GB of RAM, and an NVIDIA RTX A6000 GPU, which provides robust computational power for training KT models. The software environment includes Ubuntu 20.04 LTS as the operating system, Python 3.8, and PyTorch 1.12.0, ensuring a stable and efficient platform for implementing and testing our models.

## 4.2 EXPERIMENTAL RESULTS

To demonstrate the superiority of our method in enhancing KT through automated question groupings rather than predefined KC labeling, we conduct extensive experiments on four benchmark datasets, comparing it with eight baseline methods. Each model is evaluated in three modes: using only question information (Q), combining question and KC information (KC), and replacing KC with our automated groupings (Ours).

As illustrated in Table 2, models leveraging KCs consistently outperform those relying solely on question IDs across all datasets, emphasizing the significant contribution of KCs to improving KT

---

[1]https://sites.google.com/site/assistmentsdata/home/assistment-2009-2010-data/skill-builder-data-2009-2010

[2]https://sites.google.com/site/assistmentsdata/home/2012-13-school-data-with-affect

[3]https://pslcdatashop.web.cmu.edu/KDDCup/

Table 2: Results of the main experiments.

| Model | | ASSIST2009 | | | ASSIST2012 | | | Algebra2005 | | | Bridge2006 | | |
|---|---|---|---|---|---|---|---|---|---|---|---|---|---|
| | | AUC | ACC | RMSE | AUC | ACC | RMSE | AUC | ACC | RMSE | AUC | ACC | RMSE |
| DKT | Q | 0.7475 | 0.7177 | 0.4320 | 0.7318 | 0.7344 | 0.4240 | 0.7449 | 0.8052 | 0.4109 | 0.7422 | 0.8325 | 0.3568 |
| | KC | 0.8263 | 0.7740 | 0.3909 | 0.7388 | 0.7355 | 0.4233 | 0.8250 | 0.8160 | 0.3626 | 0.7966 | 0.8487 | 0.3349 |
| | Ours | **0.8551** | **0.7943** | **0.3778** | **0.7854** | **0.7599** | **0.4050** | **0.9169** | **0.8641** | **0.3104** | **0.8479** | **0.8541** | **0.3220** |
| DKVMN | Q | 0.7381 | 0.7082 | 0.4487 | 0.7222 | 0.7319 | 0.4266 | 0.7687 | 0.7977 | 0.3851 | 0.7586 | 0.8402 | 0.3460 |
| | KC | 0.8178 | 0.7718 | 0.3956 | 0.7356 | 0.7347 | 0.4232 | 0.8227 | 0.8149 | 0.3634 | 0.7879 | 0.8467 | 0.3371 |
| | Ours | **0.8577** | **0.7976** | **0.3762** | **0.7853** | **0.7607** | **0.4047** | **0.9146** | **0.8620** | **0.3122** | **0.8510** | **0.8547** | **0.3208** |
| AKT | Q | 0.7744 | 0.7395 | 0.4220 | 0.7197 | 0.7287 | 0.4289 | 0.7926 | 0.8020 | 0.3768 | 0.7646 | 0.8398 | 0.3457 |
| | KC | 0.8263 | 0.7738 | 0.3945 | 0.7753 | 0.7553 | 0.4094 | 0.8328 | 0.8190 | 0.3597 | 0.8084 | 0.8505 | 0.3316 |
| | Ours | **0.8642** | **0.7998** | **0.3721** | **0.7851** | **0.7604** | **0.4050** | **0.9151** | **0.8625** | **0.3116** | **0.8488** | **0.8545** | **0.3216** |
| SimpleKT | Q | 0.8086 | 0.7685 | 0.4008 | 0.7280 | 0.7339 | 0.4252 | 0.8155 | 0.8120 | 0.3669 | 0.7917 | 0.8474 | 0.3362 |
| | KC | 0.8275 | 0.7761 | 0.3953 | 0.7794 | 0.7575 | 0.4074 | 0.8386 | 0.8232 | 0.3560 | 0.8153 | 0.8526 | 0.3289 |
| | Ours | **0.8596** | **0.7940** | **0.3756** | **0.7857** | **0.7608** | **0.4047** | **0.9141** | **0.8615** | **0.3124** | **0.8520** | **0.8562** | **0.3200** |
| ReKT | Q | 0.8332 | 0.7754 | 0.3898 | 0.7663 | 0.7518 | 0.4118 | 0.8047 | 0.8107 | 0.3701 | 0.7870 | 0.8446 | 0.3386 |
| | KC | 0.8488 | 0.7882 | 0.3803 | 0.7824 | 0.7595 | 0.4056 | 0.8379 | 0.8234 | 0.3560 | 0.8152 | 0.8528 | 0.3288 |
| | Ours | **0.8679** | **0.7993** | **0.3690** | **0.7870** | **0.7611** | **0.4042** | **0.9168** | **0.8645** | **0.3095** | **0.8477** | **0.8544** | **0.3219** |
| UKT | Q | 0.8278 | 0.7735 | 0.3927 | 0.7539 | 0.7446 | 0.4169 | 0.7962 | 0.8057 | 0.3745 | 0.7848 | 0.8436 | 0.3395 |
| | KC | 0.8427 | 0.7826 | 0.3860 | 0.7790 | 0.7571 | 0.4075 | 0.8399 | 0.8234 | 0.3554 | 0.8136 | 0.8514 | 0.3296 |
| | Ours | **0.8619** | **0.7977** | **0.3741** | **0.7862** | **0.7606** | **0.4047** | **0.9148** | **0.8625** | **0.3115** | **0.8481** | **0.8545** | **0.3219** |
| FlucKT | Q | 0.8185 | 0.7694 | 0.3986 | 0.7594 | 0.7479 | 0.4145 | 0.7952 | 0.8066 | 0.3749 | 0.7864 | 0.8442 | 0.3391 |
| | KC | 0.8431 | 0.7873 | 0.3851 | 0.7850 | 0.7607 | 0.4048 | 0.8380 | 0.8201 | 0.3577 | 0.8146 | 0.8525 | 0.3289 |
| | Ours | **0.8587** | **0.7971** | **0.3759** | **0.7863** | **0.7610** | **0.4047** | **0.9143** | **0.8615** | **0.3124** | **0.8480** | **0.8536** | **0.3226** |
| LefoKT | Q | 0.8175 | 0.7684 | 0.4028 | 0.7590 | 0.7481 | 0.4148 | 0.7954 | 0.8051 | 0.3751 | 0.7868 | 0.8440 | 0.3391 |
| | KC | 0.8358 | 0.7800 | 0.3890 | 0.7835 | 0.7601 | 0.4055 | 0.8372 | 0.8209 | 0.3575 | 0.8154 | 0.8526 | 0.3287 |
| | Ours | **0.8669** | **0.8014** | **0.3703** | **0.7870** | **0.7615** | **0.4042** | **0.9163** | **0.8638** | **0.3103** | **0.8518** | **0.8553** | **0.3205** |

performance by offering more comprehensive and informative data for precise predictions. More-over, by replacing predefined KC labels with our automated groupings, all models achieve further performance improvements and surpass current state-of-the-art methods on all datasets, thereby demonstrating that our data-driven method produces more meaningful question groupings compared to KC labels. Notably, even earlier KT models like DKT and DKVMN, when enhanced with our method, outperform advanced models like ReKT and FlucKT, even though these advanced models have more complex structures. This observation suggests that the performance improvement in KT models may depend less on structural complexity and more on the quality and representation of the data. Rather than focusing on developing intricate model architectures, we should prioritize enhancing the quality of KT data, as this not only saves effort on complex model design but also offers exceptional flexibility and ease of integration, making it a practical solution for various educational applications in real-world settings.

## 4.3 HYPERPARAMETER ANALYSIS

To gain deeper insights into how different hyperparameters affect the grouping results, we conduct a series of experiments focusing mainly on two core hyperparameters: the number of groups $|\mathcal{G}|$, which determines the granularity of question partitioning, and the temperature $\lambda$, which governs the sharpness of the assignment distribution.

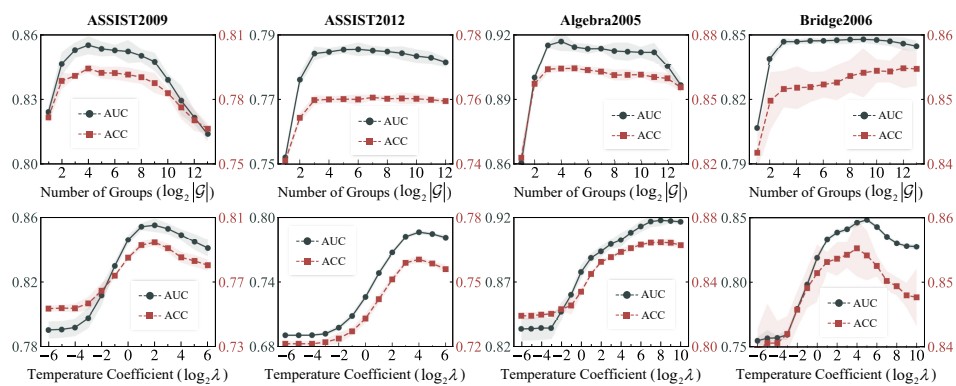

Figure 2: The impact of hyperparameter $|\mathcal{G}|$ and $\lambda$ on the experimental results.

On most datasets, as $|\mathcal{G}|$ increases, the predictive metric rises sharply at first. The model's performance improves significantly as more groups provide greater flexibility to capture question similar-

ities. However, after reaching a plateau around $|\mathcal{G}| = 2^4$, further increases in $|\mathcal{G}|$ yield diminishing returns and can even be detrimental. This is likely because very small groups fail to accumulate sufficient student-question interactions to stabilize parameter estimates. Notably, ASSIST2009, with a relatively small question pool of approximately 17K unique questions, shows a pronounced drop in performance when $|\mathcal{G}|$ becomes excessively large. An overly large $|\mathcal{G}|$ creates numerous near-empty groups with unreliable statistics, leading to rapid overfitting. Regarding the temperature $\lambda$, the performance demonstrates a clear trend of first increasing and then decreasing. When $\lambda$ is small, the assignment distribution approaches a one-hot format, causing the model to behave like a hard grouping algorithm and losing the benefits of probabilistic smoothing. As $\lambda$ increases, the distribution becomes more uniform. While this initially enhances generalization by sharing statistical strength across similar questions, excessive values of $\lambda$ erase the useful preference for low-cost groups and introduce noise, ultimately degrading performance. The optimal $\lambda$ strikes a balance, providing just the right amount of softness for the best generalization.

## 4.4 ITERATION ANALYSIS

To better understand how our method iterates and evolves in performance, we conduct a detailed analysis on the ASSIST2009 dataset, where we perform experiments using optimal configurations and record all intermediate results from the alternating minimization process. For rounds 0 to 15, we visualize the question groupings with Sankey diagrams, which track group size and evolution, illustrating how each group transforms from one round to the next. Additionally, we input results from each iteration into the KT model, completing the training-to-testing process and outputting the AUC. We also calculate the normalized mutual information (NMI) between successive rounds to reflect the grouping similarity.

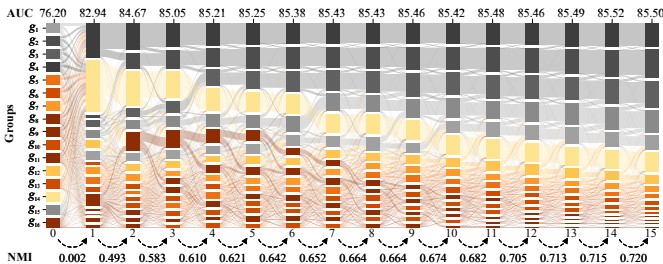

As shown in Figure 3, in the initial state, we randomly assign groups for each question, maintaining a relatively even distribution across them. In the early iterations, the AUC increases rapidly, while the NMI remains low, indicating significant fluctuations in the groupings. However, as the iterations progress, the AUC gains slow down, and the NMI rises, stabilizing around 0.7. From round 10 onward, the groupings start to stabi-

Figure 3: Iteration progress from round 0 to 15 with 16 groups on the ASSIST2009 dataset.

lize, ultimately resulting in groups of varying sizes. The experimental results indicate that the algorithm is capable of finding a stable grouping method throughout the iterations, which suggests the presence of an inherent structure within the data that our approach effectively captures.

## 4.5 ROBUSTNESS ANALYSIS

In KT tasks, the prediction accuracy of questions is significantly influenced by their frequency in the dataset. For questions that appear infrequently, models often lack sufficient training data, leading to unstable predictions that can be either too high or too low, presenting randomness. To verify whether our proposed automatic grouping method enhances the robustness of KT models, we conduct experiments on the ASSIST2009, comparing two approaches: one model is trained solely on the original data and is called the basic method, while the other is trained with grouping information and is termed our method. We plot Figure 4 with the frequency of question occurrences on the x-axis and evaluation metrics on the y-axis to compare and analyze the prediction performance of the two methods.

As shown in Figure 4, the frequency distribution of questions in the ASSIST2009 follows a long-tail pattern, with the majority of questions appearing fewer than 140 times, while the number of questions decreases as frequency increases. The distribution of the basic model for low-frequency questions is relatively dispersed, indicating significant variability in prediction stability. In contrast, our method yields a more concentrated distribution, demonstrating greater consistency and robust-

ness. This suggests that the automatic grouping method effectively compensates for the prediction instability caused by data scarcity by providing associations between questions.

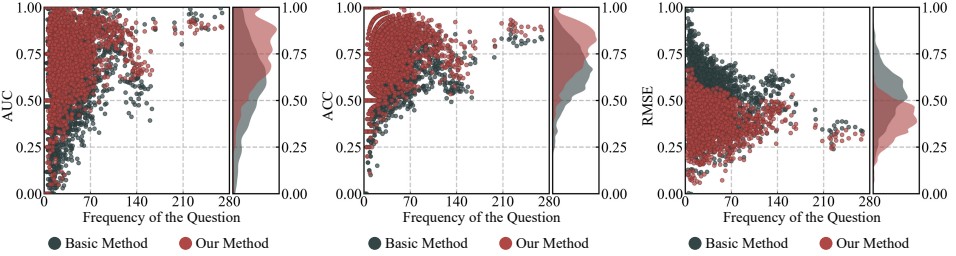

Figure 4: Comparative Analysis of Robustness: Our Method vs. Basic Method.

### 4.6 GROUPS SEMANTIC ANALYSIS

To analyze the potential semantic meaning of question groups and explore their feasibility in guiding real-world KCs' labeling, we construct a similarity matrix between groups and KCs using the group-question and KC-question relationship matrices and then derive an association matrix of KCs in the ASSIST2009 dataset. After applying a fixed threshold to eliminate edges with low relevance, we visualize the relationships in Figures 5 (a) and (b) and summarize the semantic labels of several typical groups in Figure 5 (c).

We identified seven distinct semantic themes, such as "Numbers & Proportional Reasoning," and "Algebra & Functions," among others. These themes illustrate how specific KCs can enhance KT modeling when they interact, highlighting the interconnected nature of learners' cognitive processes. Notably, we found unexpected groupings, like "Percent Of" and "Prime Number," which suggest potential hidden cognitive links that merit further investigation. Conversely, closely related concepts such as "Translations," "Reflection," and "Rotations" did not cluster together, indicating the presence of different cognitive mechanisms. Overall, our findings offer valuable insights for education and cognitive science, providing a data-driven method to uncover complex relationships between KCs.

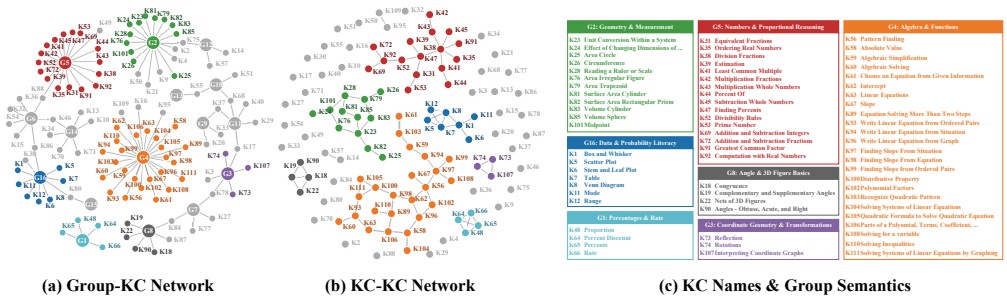

**(a) Group-KC Network**  **(b) KC-KC Network**  **(c) KC Names & Group Semantics**

Figure 5: Semantic analysis of groups and KCs.

## 5 CONCLUSION

In this work, we propose an adaptive label learning method for KT that iteratively refines question groupings through alternating optimization. Motivated by the limitations of manually annotated KC labels, our approach initializes with random groupings and alternates between optimizing the KT model parameters and updating the question groupings. By employing the relaxation operation to balance exploration and exploitation, we achieve a stable and effective grouping scheme that significantly improves the performance of various KT models. Comprehensive experiments on four real-world datasets demonstrate that our method outperforms existing baseline models, achieving state-of-the-art results. Furthermore, our approach reveals underlying semantic connections between automatic groupings and prior KCs, providing new insights into the cognitive mechanisms of learning. We believe this work offers a robust and flexible alternative to traditional KC labels, serving as a strong foundation for future KT research.

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

# A  APPENDIX

## A.1  ANALYSIS OF MONOTONIC LOSS REDUCTION

Given an iteration index $i$. The parameter space $\Theta$ is the continuous space of all trainable parameters of the KT model. Fix $\mathbf{Z} = \mathbf{Z}^{(i)}$. The loss function

$$\mathcal{L}_{\mathbf{Z}^{(i)}}(\boldsymbol{\theta}) \triangleq \mathcal{L}(\boldsymbol{\theta}, \mathbf{Z}^{(i)})$$

depends only on the model parameters $\boldsymbol{\theta}$, and $\mathcal{L}_{\mathbf{Z}^{(i)}}$ is continuously differentiable over the parameter space $\Theta \subseteq \mathbb{R}^d$. Let the current parameters be $\boldsymbol{\theta}^{(i)}$. Applying any descent-type optimization strategy to $\mathcal{L}_{\mathbf{Z}^{(i)}}$ (such as gradient descent with an appropriate step size, Adam (Kingma & Ba, 2015), etc.), the update rule ensures that

$$\mathcal{L}_{\mathbf{Z}^{(i)}}(\boldsymbol{\theta}^{(i+1)}) \leq \mathcal{L}_{\mathbf{Z}^{(i)}}(\boldsymbol{\theta}^{(i)}),$$

where $\boldsymbol{\theta}^{(i+1)}$ is the output of the algorithm. If the optimizer has converged to a local minimum, we can directly take $\boldsymbol{\theta}^{(i+1)} = \boldsymbol{\theta}^{(i)}$, in which case equality holds. Therefore, there always exists a parameter $\boldsymbol{\theta}^{(i+1)}$ satisfying

$$\mathcal{L}(\boldsymbol{\theta}^{(i+1)}, \mathbf{Z}^{(i)}) \leq \mathcal{L}(\boldsymbol{\theta}^{(i)}, \mathbf{Z}^{(i)}).$$

Fix $\boldsymbol{\theta} = \boldsymbol{\theta}^{(i+1)}$. Define the loss function as $f(\mathbf{Z}) \triangleq \mathcal{L}(\boldsymbol{\theta}^{(i+1)}, \mathbf{Z})$. The update rule for $\mathbf{Z}$ is given by:

$$\mathbf{Z}_{q,g}^{(i+1)} = \mathbb{I}\left\{g = \arg\min_{g'} \mathcal{L}_{q,g'}^{(i+1)}\right\}.$$

By construction, $\mathbf{Z}^{(i+1)}$ is chosen such that for each question $q$, it is assigned to the group $g$ that minimizes the cost $\mathcal{L}_{q,g}^{(i+1)}$. This means that for each $q$, $\mathbf{Z}_{q,g}^{(i+1)} = 1$ if $g = \arg\min_{g'} \mathcal{L}_{q,g'}^{(i+1)}$ and 0 otherwise. For the previous $\mathbf{Z}^{(i)}$, the loss is defined as:

$$f(\mathbf{Z}^{(i)}) = \mathcal{L}^{(i+1)}.$$

For the updated $\mathbf{Z}^{(i+1)}$, the loss is:

$$f(\mathbf{Z}^{(i+1)}) = \mathcal{L}^{(i+1)} + \sum_{q \in \mathcal{Q}} (\mathcal{L}_{q,\arg\min_{g'} \mathcal{L}_{q,g'}^{(i+1)}}^{(i+1)} - \mathcal{L}^{(i+1)}).$$

Here, for each question $q$, the loss difference $\mathcal{L}_{q,\arg\min_{g'} \mathcal{L}_{q,g'}^{(i+1)}}^{(i+1)} - \mathcal{L}^{(i+1)}$ is independently minimized by selecting the optimal group $g$. This independence allows us to directly sum these differences over all $q \in \mathcal{Q}$, resulting in the total loss for $\mathbf{Z}^{(i+1)}$. Therefore,

$$f(\mathbf{Z}^{(i)}) - f(\mathbf{Z}^{(i+1)}) = \sum_{q \in \mathcal{Q}} (\mathcal{L}^{(i+1)} - \mathcal{L}_{q,\arg\min_{g'} \mathcal{L}_{q,g'}^{(i+1)}}^{(i+1)}) \geq 0.$$

This is because for any $q$,

$$\mathcal{L}^{(i+1)} - \mathcal{L}_{q,\arg\min_{g'} \mathcal{L}_{q,g'}^{(i+1)}}^{(i+1)} \geq 0.$$

Thus,

$$f(\mathbf{Z}^{(i+1)}) \leq f(\mathbf{Z}^{(i)}),$$

which implies

$$\mathcal{L}(\boldsymbol{\theta}^{(i+1)}, \mathbf{Z}^{(i+1)}) \leq \mathcal{L}(\boldsymbol{\theta}^{(i+1)}, \mathbf{Z}^{(i)}).$$

In summary, by the update rule for $\mathbf{Z}$, which assigns each question to the group that minimizes the cost, we ensure that the loss does not increase. This guarantees that the sequence of losses is non-increasing, leading to convergence. This analysis provides a practical approach to understanding the convergence behavior of the alternating minimization process in our problem.

## A.2 DERIVATION OF THE SINKHORN CLOSED-FORM SOLUTION

Consider the optimization problem

$$\mathbf{P}^\star = \arg\min_{\mathbf{P} \in \Delta} \langle \mathbf{P}, \mathbf{L} \rangle + \lambda \cdot \mathrm{KL}(\mathbf{P} \| \mathbf{R}),$$

where we introduce the multipliers $\boldsymbol{\alpha} \in \mathbb{R}^{|\mathcal{Q}|}$ for the row sum constraints. The corresponding Lagrangian function is

$$\mathcal{F}(\mathbf{P}, \boldsymbol{\alpha}) = \langle \mathbf{P}, \mathbf{L} \rangle + \lambda \sum_{q,g} \mathbf{P}_{q,g} \left( \ln \frac{\mathbf{P}_{q,g}}{\mathbf{R}_{q,g}} - 1 \right) + \boldsymbol{\alpha}^\top \left( \mathbf{P} \cdot \mathbf{1}_{|\mathcal{G}|} - \mathbf{1}_{|\mathcal{Q}|} \right).$$

Taking the derivative with respect to $\mathbf{P}_{q,g}$ and setting it to zero, we have

$$\frac{\partial \mathcal{L}}{\partial \mathbf{P}_{q,g}} = \mathbf{L}_{q,g} + \lambda \ln \frac{\mathbf{P}_{q,g}}{\mathbf{R}_{q,g}} + \alpha_q = 0,$$

which yields

$$\mathbf{P}_{q,g} = \mathbf{R}_{q,g} \exp\left( -\frac{\mathbf{L}_{q,g} + \alpha_q}{\lambda} \right).$$

Since

$$\sum_g \mathbf{P}_{q,g} = \sum_g \mathbf{R}_{q,g} \exp\left( -\frac{\mathbf{L}_{q,g} + \alpha_q}{\lambda} \right) = 1,$$

we define

$$\mathbf{K}_{q,g} = \mathbf{R}_{q,g} \exp\left(-\frac{\mathbf{L}_{q,g}}{\lambda}\right), \qquad u_q = \exp\left(-\frac{\alpha_q}{\lambda}\right).$$

Thus, the compact form of the above equation can be written as

$$\mathbf{P} = \text{diag}(\mathbf{u}) \cdot \mathbf{K}, \quad \text{diag}(\mathbf{u}) \cdot \mathbf{K} \cdot \mathbf{1}_{|\mathcal{G}|} = \mathbf{1}_{|\mathcal{Q}|}.$$

To further enforce column sum normalization, i.e., $\mathbf{P}^\top \mathbf{1}_{|\mathcal{Q}|} = \mathbf{1}_{|\mathcal{G}|}$, we introduce a second set of multipliers $\beta$ and define

$$v_g = \exp\left(-\frac{\beta_g}{\lambda}\right),$$

resulting in

$$\mathbf{P} = \text{diag}(\mathbf{u}) \cdot \mathbf{K} \cdot \text{diag}(\mathbf{v}).$$

The pair $(\mathbf{u}, \mathbf{v})$ is then updated via the Sinkhorn iteration

$$u_q \leftarrow \left(\sum_g \mathbf{K}_{q,g} v_g\right)^{-1}, \qquad v_g \leftarrow \left(\sum_q u_q \mathbf{K}_{q,g}\right)^{-1},$$

alternating until both $\mathbf{P1} = \mathbf{1}$ and $\mathbf{P}^\top \mathbf{1} = \mathbf{1}$ hold. The final converged solution is

$$\mathbf{P}^\star = \text{diag}(\mathbf{u}) \cdot \mathbf{K} \cdot \text{diag}(\mathbf{v}), \quad \mathbf{K}_{q,g} = \mathbf{R}_{q,g} \exp\left(-\frac{\mathbf{L}_{q,g}}{\lambda}\right).$$

### A.3 DETAILED ANALYSIS OF ALGORITHMIC COMPLEXITY

In the first stage of alternating minimization, we fix the question assignment matrix $\mathbf{Z}$ and perform a full model training for the continuous variable $\boldsymbol{\theta}$. Assuming we use mini-batch gradient descent with a batch size of $B$, the time complexity for one forward and backward pass is approximately

$$\mathcal{O}(B \cdot T_{\text{avg}} \cdot D^2),$$

where $D^2$ arises from the multiplication of hidden states and weight matrices in structures such as RNN/LSTM/Transformer. Training one epoch requires iterating over $|\mathcal{U}|$ samples, with a total of $|\mathcal{U}|/B$ batches. Therefore, the overall complexity for one $\boldsymbol{\theta}$ update stage is

$$\mathcal{O}\left(E \cdot \frac{|\mathcal{U}|}{B} \cdot B \cdot T_{\text{avg}} \cdot D^2\right) = \mathcal{O}\left(E \cdot |\mathcal{U}| \cdot T_{\text{avg}} \cdot D^2\right).$$

In the second stage, we need to compute the potential loss $\mathcal{L}_{q,g}$ for each question $q$ and each group $g$, i.e., construct the loss matrix $\mathbf{L}$. Calculating the loss for the same $g$ across all $q$ involves one forward pass, with a time complexity of approximately $\mathcal{O}(|\mathcal{U}| \cdot T_{\text{avg}} \cdot D^2)$. Therefore, the overall complexity for constructing the entire loss matrix is

$$\mathcal{O}\left(|\mathcal{G}| \cdot |\mathcal{U}| \cdot T_{\text{avg}} \cdot D^2\right).$$

In the Sinkhorn iteration, $\mathbf{G} \in \mathbb{R}^{|\mathcal{Q}| \times |\mathcal{G}|}$. The complexity of each iteration is $\mathcal{O}(|\mathcal{Q}| \cdot |\mathcal{G}|)$. If the iteration converges after $S$ rounds, the total complexity is

$$\mathcal{O}(S \cdot |\mathcal{Q}| \cdot |\mathcal{G}|).$$

Since $S$ is typically small, this part of the overhead is relatively low. Obtaining the hard assignment $\mathbf{Z}$ from the probability matrix $\mathbf{P}$ via $\arg\max$ requires comparing $|\mathcal{G}|$ values for each row, with a time complexity of

$$\mathcal{O}(|\mathcal{Q}| \cdot |\mathcal{G}|),$$

which is negligible in the overall overhead.

In one outer iteration of alternating minimization, the main overhead comes from the model training stage and the loss matrix computation stage. Therefore, the total complexity for a single iteration is

$$\mathcal{O}\left(E \cdot |\mathcal{U}| \cdot T_{\text{avg}} \cdot D^2 + |\mathcal{G}| \cdot |\mathcal{U}| \cdot T_{\text{avg}} \cdot D^2 + S \cdot |\mathcal{Q}| \cdot |\mathcal{G}|\right).$$

After executing $I_{\max}$ iterations, the overall complexity is

$$\mathcal{O}\left(I_{\max} \cdot \left[E \cdot |\mathcal{U}| \cdot T_{\text{avg}} \cdot D^2 + |\mathcal{G}| \cdot |\mathcal{U}| \cdot T_{\text{avg}} \cdot D^2 + S \cdot |\mathcal{Q}| \cdot |\mathcal{G}|\right]\right).$$

## A.4 DETAILED INTRODUCTION FOR BASELINE METHODS

We provide a comprehensive overview of the baseline methods used for comparison in our experiments. These methods are widely recognized in the field of KT and serve as a solid foundation for evaluating the performance of our proposed model. The selected baselines include:

**DKT** (Piech et al., 2015) is the first deep learning-based KT model that employs an LSTM layer to encode students' knowledge states for predicting their future responses.

**DKVMN** (Zhang et al., 2017) introduces a dynamic key-value memory network to model the relationships between concepts and directly output a student's mastery level of each concept.

**AKT** (Ghosh et al., 2020) couples attention-based neural networks with cognitive and psychometric models, offering a balance between flexibility and interpretability in KT.

**SimpleKT** (Liu et al., 2023b) provides a lightweight yet effective baseline for KT using the Rasch model to explicitly capture question-specific variations and dot-product attention for time-aware information.

**ReKT** (Shen et al., 2024) proposes a Forget-Response-Update (FRU) framework inspired by human cognitive development models, achieving high performance with minimal computational resources.

**UKT** (Cheng et al., 2025) introduces stochastic embeddings and a Wasserstein self-attention mechanism to model uncertainty in student interactions and enhance robustness.

**FlucKT** (Hou et al., 2025) enhances attention mechanisms by integrating causal convolution and kernelized bias to better capture short-term cognitive fluctuations.

**LefoKT** (Bai et al., 2025) introduces relative forgetting attention to decouple forgetting patterns from problem relevance, improving attention-based models' capability to handle continuous forgetting processes.

