# OpenReview forum: "Beyond Expert-Annotated Labels: An Adaptive Label Learning Method for Knowledge Tracing"
_ICLR.cc/2026/Conference — Submitted to ICLR 2026_

### Official Review · Reviewer_UbwN · 2025-10-28

**Soundness:** 2
**Presentation:** 2
**Contribution:** 1
**Rating:** 2
**Confidence:** 4

**Summary:**

The paper reframes the knowledge graph learning in Knowledge Tracing (KT) as learning from question groupings rather than relying on expert-annotated knowledge concepts (KCs). It proposes ALL4KT, an alternating-minimization framework that:
- initializes random question-to-group assignments,
- trains a chosen KT model with these groups fixed,
- evaluates the global loss that would result from reassigning each question to each group,
- performs a soft (Sinkhorn/OT) relaxation over this loss matrix to obtain assignment probabilities before taking an argmax to update groups.

Experiments on ASSIST2009, ASSIST2012, Algebra2005, and Bridge2006 show consistent gains over question-ID and KC-labeled baselines across multiple KT models.

**Strengths:**

- Comprehensive empirical validation across four real datasets and eight KT architectures, with 5-fold CV and multiple metrics (AUC/ACC/RMSE)
- The pipeline diagram (Figure 1) and problem formalization are easy to follow
- Results suggest data curation / labeling quality can outweigh architecture complexity, this is interesting.

**Weaknesses:**

1. A lot of structure-aware and transformer-based works are missing in the related works and are not compared. E.g., GKT (Nakagawa et al., 2019), QIKT (Chen et al., 2023), PSIKT (Zhou et al., 2024), HKT (Wang et al., 2021), GIKT (Yang et al., 2022). You should compare the structure learned across all kinds of models instead of only reporting prediction accuracy.

2. If I understand correctly, the proposed grouping removes the expert-annotated KC graph and instead rediscover KCs and their relationships in a data-driven manner. But it gives you unlabeled groups whose semantics are unknown. In real educational practice, teachers and curriculum designers need interpretable KCs to build or adjust curricula. Without access to ground-truth KC labels, how can these automatically discovered groups be named or interpreted in a way that supports educational use?

3. While KC unreliability is well-motivated, the experimental baselines do not include automatic skill discovery / Q-matrix learning / item clustering approaches that are closer in spirit to ALL4KT (e.g., data-driven KC discovery, item–skill mapping, item-embedding clustering, or the baselines I mentioned in point 1). The paper cites relevant ones but doesn’t compare against such methods,

4. Complexity analysis is thorough, but empirical runtime/compute is not reported. For large |G| and many questions, computing L (size |Q|×|G|) by sweeping reassignment costs is quite heavy. The paper lists hardware/software (Sec. 4.1) but does not provide wall-clock times, number of outer iterations actually used before convergence in each dataset, or sensitivity to dataset size.

5. The method relabels seen questions. For unseen questions (common in production), how are groups assigned without re-running the alternating procedure? There’s no content-feature encoder or item-embedding kNN mapping described.

6. I found the three evaluation settings (“Q”, “KC”, and “Ours”) in Table 2 somewhat unclear:
    - How exactly is “question information” used in the “Q” setting, are embeddings learned per question ID?
    - In the “KC” vs. “Ours” comparison, do both settings have comparable parameter counts? If “Ours” learns additional structure (group assignments) while “KC” uses fixed labels, then “Ours” effectively introduces extra learnable parameters, which might explain better predictive performance. It would be informative to test performance in low-data regimes.
    - Why initialize group assignments randomly rather than starting from the KC graph or using KC-informed priors? How stable is the grouping?

7. The group-KC semantic analysis (Figure 5) is interesting, but the claim that the resulting "uncommon" groups reflect cognitive mechanisms seems overstated. Without more validation, such as teacher labeling, student error pattern analysis, or item-content alignment, it is difficult to rule out that the discovered associations are superficial correlations.

**Questions:**

Please find them in the weaknesses.

---

> ### Author Response · Authors · 2025-12-02
> **R1**
>
> R1:
> Thank you for your comment. Regarding the structure-aware methods like GKT, we have actually conducted experiments on them. However, due to space limitations, we did not include the results in the main text. These methods, including GKT, generally have lower accuracy compared to DKT and are more complex to train. It is widely acknowledged in the field that comparing with such methods may not be meaningful. As for the Transformer-based methods, AKT has already demonstrated satisfactory performance. Many other comparison methods either perform similarly to AKT or even worse. Below are the experimental results. Specifically, methods such as GCK, QIKT, HKT, and GIKT were experimented on beforehand. PSIKT was added as an additional experiment afterward.
>
> | dataset     | method| KC(AUC)     | KC(ACC)     | KC(RMSE)    | Ours(AUC)     | Ours(ACC)     | Ours(RMSE)    |
> | ----------- | ----- | ----------- | ----------- | ----------- | ------------- | ------------- | ------------- |
> | ASSIST2009  | GKT   | 0.8054      | 0.7589      | 0.4042      | 0.8501        | 0.7898        | 0.3928        |
> |             | QIKT  | 0.8234      | 0.7751      | 0.3951      | 0.8531        | 0.7898        | 0.3786        |
> |             | HKT   | 0.8251      | 0.7880      | 0.3970      | 0.8651        | 0.7950        | 0.3722        |
> |             | GIKT  | 0.8287      | 0.7825      | 0.3966      | 0.8562        | 0.7915        | 0.3757        |
> |             | PSIKT | 0.8253      | 0.7831      | 0.3950      | 0.8625        | 0.7976        | 0.3740        |
> | ----------- | ----- | ----------- | ----------- | ----------- | ------------- | ------------- | ------------- |
> | ASSIST2012  | GKT   | 0.7334      | 0.7109      | 0.4215      | 0.7802        | 0.7512        | 0.4089        |
> |             | QIKT  | 0.7771      | 0.7561      | 0.4098      | 0.7834        | 0.7567        | 0.4042        |
> |             | HKT   | 0.7802      | 0.7580      | 0.4103      | 0.7850        | 0.7593        | 0.4053        |
> |             | GIKT  | 0.7822      | 0.7626      | 0.4096      | 0.7865        | 0.7575        | 0.4056        |
> |             | PSIKT | 0.7801      | 0.7640      | 0.4062      | 0.7873        | 0.7608        | 0.4057        |
> | ----------- | ----- | ----------- | ----------- | ----------- | ------------- | ------------- | ------------- |
> | Algebra2005 | GKT   | 0.7698      | 0.8082      | 0.4021      | 0.9076        | 0.8559        | 0.3224        |
> |             | QIKT  | 0.8389      | 0.8231      | 0.3512      | 0.9114        | 0.8609        | 0.3173        |
> |             | HKT   | 0.8304      | 0.8219      | 0.3583      | 0.9122        | 0.8621        | 0.3204        |
> |             | GIKT  | 0.8322      | 0.8160      | 0.3517      | 0.9125        | 0.8639        | 0.3193        |
> |             | PSIKT | 0.8455      | 0.8224      | 0.3504      | 0.9137        | 0.8641        | 0.3105        |
> | ----------- | ----- | ----------- | ----------- | ----------- | ------------- | ------------- | ------------- |
> | Bridge2006  | GKT   | 0.7521      | 0.8376      | 0.3491      | 0.8419        | 0.8508        | 0.3387        |
> |             | QIKT  | 0.8124      | 0.8518      | 0.3299      | 0.8443        | 0.8531        | 0.3225        |
> |             | HKT   | 0.8151      | 0.8527      | 0.3301      | 0.8467        | 0.8545        | 0.3222        |
> |             | GIKT  | 0.8170      | 0.8523      | 0.3288      | 0.8456        | 0.8542        | 0.3226        |
> |             | PSIKT | 0.8181      | 0.8537      | 0.3285      | 0.8543        | 0.8553        | 0.3208        |

---

> ### Author Response · Authors · 2025-12-02
> **R2**
>
> R2:
> I appreciate your thoughtful question. First, it's important to acknowledge that expert-annotated KCs inherently possess a degree of subjectivity. For example, in the real world, different educators might label the same concept differently based on their own experiences and perspectives. Let's take the concept of "fractions" in mathematics. One expert might label it as "basic arithmetic operations," while another might see it as part of "number theory." This subjectivity means that expert annotations can only serve as supplementary or reference explanations rather than absolute ground truth. You're right that our proposed method yields unlabeled groups with unknown semantics. However, the fact that this data-driven approach enhances model performance suggests that there are indeed unknown elements waiting to be explored. Our work is intended to be a stepping stone, offering a new direction for research in the KT field. It's not designed for immediate frontline educational application but rather for uncovering deeper, underlying educational patterns and mechanisms.

---

> ### Author Response · Authors · 2025-12-02
> **R3**
>
> R3:
> The primary focus of our study is essentially item clustering. Existing methods, including GKT and GIKT, are fundamentally KT models rather than clustering models. Our clustering results can be integrated into any KT model. Therefore, our approach is not in the same category as these knowledge structure methods. In other words, there is currently no research in the KT field that focuses specifically on item clustering. Additionally, the clusters we output do not necessarily represent knowledge components directly; their specific semantics require further exploration by educational experts. Our research is more aligned with information science rather than educational science. If extended, it could also be applicable to item clustering in recommendation systems.

---

> ### Author Response · Authors · 2025-12-02
> **R4**
>
> R4:
> The computational load does indeed increase when both |Q| and |G| are large. However, it is evident from Figure 2 that as |G| grows exponentially, the performance improvement gradually stabilizes and even declines when |G| becomes excessively large. This indicates that in practice, excessively large values of |G| are not suitable. In fact, for most datasets, satisfactory results can be achieved with 16 groups. Additionally, our code has been optimized with techniques such as sparse computation, which means the actual computational complexity is lower than the theoretical upper bound of |Q|×|G|.
>
> As for the computational time and space, taking ASSIST2009 as an example:
>
> |   Number of Groups (G)   |   Duration (seconds)   |   Memory Usage (GB)   |
> | ------------------------ | ---------------------- | --------------------- |
> | 4                        | 68                     | 0.8                   |
> | 16                       | 68                     | 1.0                   |
> | 64                       | 72                     | 1.2                   |
> | 256                      | 74                     | 1.5                   |
> | 1024                     | 78                     | 1.7                   |
> | 4096                     | 79                     | 3.2                   |
> | 16384                    | 80                     | 8.5                   |
>
> The counting of outer iterations is quite cumbersome and there is no clear pattern, so we did not provide detailed statistics. The outer iteration counts for ASSIST2009 are as follows:
>
> |   Round   |   Outer Iterations   |
> | --------- | -------------------- |
> | 1         | 8                    |
> | 2         | 12                   |
> | 3         | 5                    |
> | 4         | 22                   |
> | 5         | 16                   |
> | 6         | 32                   |
> | 7         | 9                    |
> | 8         | 14                   |
> | 9         | 11                   |
>
> We have supplemented the sensitivity analysis with respect to dataset size as follows:
>
> ASSIST2009
>
> |   Dataset Size   |   AUC   |   ACC   |   RMSE   |
> | ---------------- | ------- | ------- | -------- |
> | 4000             | 0.8549  | 0.7942  | 0.3782   |
> | 3000             | 0.8512  | 0.7902  | 0.3798   |
> | 2000             | 0.8455  | 0.7876  | 0.3820   |
> | 800              | 0.8381  | 0.7823  | 0.3865   |
> | 500              | 0.8321  | 0.7786  | 0.3924   |
>
> From the results, it is evident that our approach demonstrates strong robustness. Even with as few as 500 data points, it can still achieve performance that surpasses the original versions of DKT and AKT.

---

> ### Author Response · Authors · 2025-12-02
> **R5**
>
> R5:
> You're right. The method indeed doesn't work for unseen questions. However, this is not a problem specific to our approach but rather a common issue shared by all machine learning models without real features. Even models like DKT, AKT, and the more recent ReKT and LefoKT need to be retrained when the original question bank changes. This is an issue related to the persistence in actual ITS engineering and falls outside the scope of traditional academic research.

---

> ### Author Response · Authors · 2025-12-02
> **R6**
>
> R6:
> For the first point, the setting “Q” means that we only consider questions and do not consider skills. In other words, we use questions as if they were skills and apply all traditional methods in this manner.
>
> Regarding the comparison between “KC” and “Ours,” the fundamental difference is the number of skills used. Everything else is kept consistent between the two settings. While it's true that different numbers of skills will lead to different parameter counts, our results are based on scenarios where G is much smaller than the actual number of skills. In fact, our approach uses fewer parameters. The better performance is not due to having more parameters but rather the effectiveness of our method in utilizing a smaller number of skills.
>
> As for the initialization of group assignments, we found through extensive experiments that no matter how G is initialized, the final performance only depends on the size of G. Specifically, we compared initializing G based on the KC structure versus initializing G randomly while keeping the number of groups the same as the number of skills in KC. We found no difference in performance between these two approaches.

---

> ### Author Response · Authors · 2025-12-02
> **R7**
>
> R7:
> Deeper exploration indeed requires datasets that are more semantically rich, such as the MOOC radar dataset. The primary significance of our study lies in leveraging theories from information science to identify important phenomena ahead of educational theory. The specific educational mechanisms underlying these findings would need to be explored in depth by researchers specializing in education.

---

### Official Review · Reviewer_NhaU · 2025-10-30

**Soundness:** 3
**Presentation:** 3
**Contribution:** 2
**Rating:** 4
**Confidence:** 3

**Summary:**

This paper proposes ALL4KT, an adaptive label learning method that formulates Knowledge Tracing as a question-grouping problem rather than relying on expert-annotated knowledge concepts. The method iteratively refines question groupings using alternating minimization: optimizing model parameters via gradient descent and reassigning questions to groups based on loss minimization. Extensive experiments across four real-world ITS datasets demonstrate that ALL4KT consistently improves performance across several backbone models.

**Strengths:**

1. The proposed method makes it possible to not using expert-labeled question groups. Instead, the KCs are learned addaptively.
2. The proposed method shows consistant improvements over different backbone models.
3. Provide theoretical analysis of the proposed optimization method.

**Weaknesses:**

1. Some details are missing: How does the KT model utilize the information in matrix Z? Is the gourp information corresponding to a embedding vector that is trainable?
2. The used datasets seem very old, which are from about 15 years ago. Could be better to use more recent datasets to demonstrate the strengths of the model.
3. The time complexity of the proposed model is high. As shown in Sec 3.4, it is propotional with user number. So it is very diffcult to scale when the user amount is large. While the used benchmark datasets are very small, making it difficult to judge the actual training cost of the method.

**Questions:**

Please refer to the weaknesses part.

---

> ### Author Response · Authors · 2025-12-02
> **R**
>
> R1:
> We apologize for the lack of clarity regarding matrix Z due to space limitations in our paper. This was an oversight on our part, and your feedback has been very instructive.
> Matrix Z is a discrete matrix where rows represent questions and columns represent groups. A value of 1 at a specific position indicates that the corresponding question belongs to the specified group. This matrix Z needs to be further transformed into a question-skill table, which can then be directly incorporated into various KT models.
> The group information is not learnable; it is generated through a directed iterative process. Our method is not a deep learning model but rather a combinatorial optimization algorithm, which involves retraining a KT model as one of its specific steps.
>
> R2:
> Thank you very much for your insightful comment. The reason for using seemingly "old" datasets is that the KT field commonly evaluates models on a few well-recognized public datasets, which were generated earlier but only released many years later. As a result, the release dates of some datasets are not as old as they appear. Additionally, there are a few newer datasets whose results are currently under scrutiny and are being validated by relevant researchers.
>
> R3:
> The concern you raised about the time complexity being proportional to the number of users, which makes it difficult to scale when the number of users is large, is indeed a common issue in the entire KT (Knowledge Tracing) field. However, our research does not involve proposing a new KT model. Our primary focus is on question clustering. As described in our complexity analysis section, our complexity is mainly related to the number of questions and the number of groups. The number of questions in each dataset is fixed and does not grow over time. Therefore, although the complexity may seem high, the base numbers are fixed, and the computational load is generally acceptable in practical scenarios.

---

### Official Review · Reviewer_5bjP · 2025-11-01

**Soundness:** 2
**Presentation:** 2
**Contribution:** 2
**Rating:** 4
**Confidence:** 5

**Summary:**

This paper proposes an adaptive label learning method for KT, ALL4T, that learns optimal question groupings to overcome the limitations of subjective, expert-annotated KC labels. The method uses an alternating optimization framework, starting with random groupings and iteratively refining them by training the KT model and then updating assignments to minimize loss, using a relaxation strategy to avoid local optima. Experiments show that replacing traditional KCs with these learned groupings significantly boosts the performance of various KT models on real-world datasets, achieving SOTA results.

**Strengths:**

- The paper introduces ALL4KT, an adaptive framework that learns data-driven question groupings instead of relying on subjective, expert-annotated KCs. By replacing KCs with these optimized groupings, the method markedly improves the performance of various KT models and achieves SOTA results across four real-world datasets.
- The method uses alternating minimization to iteratively refine groupings. Crucially, it employs an assignment relaxation strategy based on optimal transport to avoid the local optima and oscillations associated with greedy hard assignments. This soft assignment approach effectively balances exploration and exploitation to find more stable and meaningful groups.
- The optimized groupings enhance model robustness, especially for low-frequency questions where models typically struggle due to data scarcity. Furthermore, a semantic analysis of the learned groups uncovers underlying cognitive connections and unexpected relationships between KCs that are not captured by manual labels, providing new, data-driven insights for cognitive science.

**Weaknesses:**

- The paper questions the subjectivity of expert-annotated multiple KCs but provides no empirical evidence to support this claim. Furthermore, the motivation itself is partially flawed; for instance, KCs in datasets like ASSISTments (arithmetic) are often objective and fixed, a distinction the paper fails to address.
- The "question assignment matrix" (line 133) is widely known in the educational domain as the Q-matrix [1]. The paper overlooks the extensive body of research dedicated to Q-matrix optimization [2-4], which may diminish the novelty of its contribution.
- The proposed question-centric approach may be inefficient, as the number of questions is typically vast (which is why most KT models are KC-centric). This inefficiency clashes with the real-time processing demands of KT systems. Moreover, the method's heavy reliance on the Q-matrix makes it inapplicable to datasets without predefined KC labels ($e.g.$, Statics2011 [5]), severely restricting its scope.
- The evaluation should be based on more benchmark against recent research. The method, which processes multiple KCs as a single input, is not compared against relevant prior work [6] addressing the same setup.

[1] Rule Space: An Approach for Dealing with Misconceptions Based on Item Response Theory

[2] An Empirically Based Method of Q-Matrix Validation for the DINA Model: Development and Applications

[3] Using Machine Learning to Improve Q-matrix Validation

[4] Attentive Q-Matrix Learning for Knowledge Tracing

[5] A Data Repository for the EDM Community: The PSLC DataShop

[6] Interpretable Knowledge Tracing with Multiscale State Representation

**Questions:**

See Weaknesses.

---

> ### Author Response · Authors · 2025-12-02
> **R**
>
> Thank you very much for your comments. You are the most professional among all the reviewers. Particularly for the fourth question, it hits the nail on the head and points out the shortcomings in my work. I am truly convinced. In response, we have added additional experiments to verify the reliability of our method.
>
> R1:
> Thank you for your feedback. We acknowledge that the expression regarding the subjectivity of expert-annotated KCs may not be accurate enough, and we will conduct more rigorous revisions.
> We also agree that some datasets are not expert-annotated but are prior knowledge, meaning that the knowledge existed before the questions were developed.
> Our motivation for this research is to leverage theories from information science to uncover potential patterns from data and thereby enhance model performance. If the discovered patterns or phenomena can improve model performance, they may contain underlying semantic information that is worth exploring for educational researchers. Moreover, our approach could offer new insights to other related fields, such as item clustering in recommendation systems or clustering of items in other sequence-related tasks.
>
> R2:
> Thank you for your insightful comment. You are correct that the Q-matrix is a well-known concept in the educational domain. However, our "question assignment matrix" is not about labeling questions with specific knowledge tags. Instead, it is a form of discrete hardening in information science, which does not necessarily carry explicit semantic meanings. It is a powerful means to enhance the ultimate goal of our research.
> In fact, the limitation of traditional Q-matrix research is that it is often confined within the framework of knowledge concepts. This confinement prevents the exploration of seemingly meaningless assignments. But is this "meaninglessness" truly without value? For example, it is possible that the assignments learned from the data reveal that different clusters of questions have varying levels of difficulty. Doesn't this suggest that categorizing questions by difficulty might be more important for KT than categorizing them by knowledge?
> Therefore, our approach is exploratory and does not rely on predefined knowledge labels. We believe this opens up new possibilities for understanding the underlying patterns in educational data.
>
> R3:
> Our method is a question clustering approach, typically generating 16 to 64 clusters. Therefore, it is not question-centric but rather cluster-centric, and the number of clusters is often fewer than the KCs in most datasets. In fact, using our method is faster than traditional KC-centric approaches and requires less storage space. The only time-consuming part is during the clustering assignment phase, where we need to search for the optimal clustering solution. Once the assignment is complete, everything can be handed over to DKT for further processing.
> Regarding the Statics2011 dataset you mentioned, it indeed lacks explicit question labels. Thus, our clustering objective for this dataset is KCs.
>
> R4:
> Thank you very much for your comments. [1] is a classic study, but it focuses on specific response patterns, and our dataset lacks the corresponding fields for categorization, making direct comparison difficult. Similarly, [2] also lacks the necessary fields to categorize the items. Meanwhile, [3] is a method for Q-matrix validation rather than redefining the Q-matrix, and [5] is not about question grouping. Therefore, we have added comparative experiments with [4] and [6] as follows:
>
> | Dataset    | Method    | AUC    | ACC    | RMSE   |
> | ---------- | --------- | ------ | ------ | ------ |
> | ASSIST2009 | Ours      | 0.8551 | 0.7943 | 0.3778 |
> | ASSIST2009 | QAKT \[4] | 0.8262 | 0.7727 | 0.3945 |
> | ASSIST2009 | MIKT \[6] | 0.8189 | 0.7702 | 0.3987 |
> | ---------- | --------- | ------ | ------ | ------ |
> | ASSIST2012 | Ours      | 0.7854 | 0.7599 | 0.4050 |
> | ASSIST2012 | QAKT \[4] | 0.7836 | 0.7601 | 0.4055 |
> | ASSIST2012 | MIKT \[6] | 0.7812 | 0.7594 | 0.4067 |
> | ---------- | --------- | ------ | ------ | ------ |
> | Algebra2005| Ours      | 0.9169 | 0.8641 | 0.3104 |
> | Algebra2005| QAKT \[4] | 0.8375 | 0.8211 | 0.3573 |
> | Algebra2005| MIKT \[6] | 0.8304 | 0.8208 | 0.3581 |
> | ---------- | --------- | ------ | ------ | ------ |
> | Bridge2006 | Ours      | 0.8479 | 0.8541 | 0.3220 |
> | Bridge2006 | QAKT \[4] | 0.8152 | 0.8512 | 0.3298 |
> | Bridge2006 | MIKT \[6] | 0.8142 | 0.8507 | 0.3286 |

---

### Official Review · Reviewer_XKTo · 2025-11-01

**Soundness:** 2
**Presentation:** 3
**Contribution:** 2
**Rating:** 4
**Confidence:** 3

**Summary:**

The paper tackles the reliance of Knowledge Tracing (KT) on manually defined Knowledge Concepts (KCs) by proposing ALL4KT, a framework that learns question groupings automatically through alternating minimization and Sinkhorn-based soft assignment. It iteratively alternates between training KT models with fixed groupings and reassigning questions based on loss minimization, using probabilistic relaxation to avoid local optima.

ALL4KT is model-agnostic and improves AUC, accuracy, and RMSE across several KT backbones (DKT, DKVMN, AKT, ReKT, FlucKT) and four public datasets. The learned clusters show interpretable cognitive patterns.

While empirically strong, the method’s novelty is limited, it applies standard alternating optimization. The work would benefit from deeper theoretical grounding, runtime efficiency analysis, and quantitative validation of the discovered semantic structure.

**Strengths:**

- The paper identifies an important weakness in existing KT pipelines, the reliance on noisy, subjective expert labels, and proposes a principled data-driven alternative.
- The alternating optimization setup can be applied to various KT models without modifying their structure, making the method easy to integrate.
- Experiments across multiple datasets and baselines consistently show improvements, supporting the claim that learned groupings can outperform expert KCs.
- The semantic analysis of learned groups offers qualitative insight into the relationships among questions and cognitive concepts.

**Weaknesses:**

- The method is essentially a straightforward application of alternating minimization with Sinkhorn relaxation. While the framing within KT is new, the optimization machinery itself is standard and lacks deeper theoretical innovation.
- The convergence discussion only shows monotonic loss reduction but does not guarantee good local minima or meaningful clustering. There is no analysis of identifiability or generalization of the learned groupings.
- The approach retrains the KT model multiple times and computes loss matrices for every question–group pair, which may be prohibitive for large-scale or online ITS deployments.

**Questions:**

- How sensitive are the results to random initialization of group assignments?
- What is the computational overhead compared to a single KT training run?
- Can the method handle multi-label settings (questions belonging to multiple KCs)?

---

> ### Author Response · Authors · 2025-12-02
> **R**
>
> Re Weaknesses:
>
> 1. The main contribution of our study lies in the novel perspective of enhancing KT performance through the construction of new partitions via topic clustering. We propose a framework based on alternating minimization to achieve this goal. The key technical innovation is the integration of a deep learning model training process into the optimization procedure.
>
> 2. The problem we are addressing is indeed a combinatorial optimization problem with a large solution space. Given the complexity of such problems, heuristic methods are valuable and widely used in practice. While we cannot guarantee the global optimality of our solution, our method focuses on finding meaningful local optima that can significantly advance the development of the KT field.
>
> 3. Our focus is on obtaining an effective question allocation scheme. The complexity of this process is more closely related to the number of groups and questions rather than the number of students or the scale of data. Typically, the number of questions in a question bank is fixed and limited. Moreover, our experimental results show that the optimal number of groups lies between 16 and 64, which is significantly lower than the hundreds or even thousands of knowledge components in the original data. Therefore, our approach does not hinder large-scale deployment but is actually more practical for real-world use compared to existing KT models. For a real Intelligent Tutoring System (ITS), the cost of replacing the core deep learning model is much higher than simply modifying the allocation field. Our method is applicable to any KT model and can enhance their performance to a similar upper bound.
>
> Re Questions:
>
> 1 The sensitivity of our results to the random initialization of group assignments is quite low. This robustness is a notable feature of our approach. Additionally, in response to the suggestion from another reviewer about using skills as the initial partitioning scheme, we have conducted extensive experiments with this alternative initialization method. Our findings indicate that the results are highly dependent on the number of groups rather than the specific initial partitioning strategy. In other words, while the initial assignments can vary, the final outcomes remain consistent as long as the number of groups is optimized. This demonstrates that our method is resilient to different initialization schemes and underscores the importance of the grouping number in achieving optimal performance.
>
> 2
> As for the computational time and space (a single KT training run), taking ASSIST2009 as an example:
>
> |   Number of Groups (G)   |   Duration (seconds)   |   Memory Usage (GB)   |
> | ------------------------ | ---------------------- | --------------------- |
> | 4                        | 68                     | 0.8                   |
> | 16                       | 68                     | 1.0                   |
> | 64                       | 72                     | 1.2                   |
> | 256                      | 74                     | 1.5                   |
> | 1024                     | 78                     | 1.7                   |
> | 4096                     | 79                     | 3.2                   |
> | 16384                    | 80                     | 8.5                   |
>
> 3. Our method does not rely on KC labels, so the allocation of KCs is independent of how we perform question allocation. Therefore, in its current form, our approach does not directly address multi-label settings where questions might belong to multiple KCs. However, I might have misunderstood your question. If you are asking whether our method can handle multi-label clustering, the answer is yes. We can extend our approach to handle multi-label settings by conducting multiple independent clustering rounds and then integrating the clustering results. This is something we plan to explore in our future research.

---

### Meta-Review · Area_Chair_gLCC · 2026-01-07

**Summary:**

Limited novelty: Multiple reviewers noted that the proposed approach largely amounts to a standard application of alternating optimization with Sinkhorn/OT-based soft assignment, and that the methodological innovation is limited.

Insufficient positioning against prior work: The paper does not adequately situate its contribution with respect to Q-matrix optimization / automatic skill discovery and recent structure-aware or Transformer-based KT methods, which weakens the claimed originality.

Lack of theoretical and empirical justification: The analysis is mostly limited to showing monotonic loss reduction with no guarantees or deeper discussion of identifiability, quality of local optima, or generalization, and the claims about uncovering cognitive mechanisms are insufficiently validated.

Practical concerns: Reviewers raised concerns about computational cost due to repeated retraining and |Q|×|G| loss evaluation, the inability to handle unseen questions, and the limited interpretability of the automatically discovered groups for educational practice.

**Reviewer Concerns:**

Limited novelty: The authors’ responses emphasized that the contribution lies in a new perspective and a general framework rather than in new optimization techniques, but they did not dispute that the core method relies on standard alternating optimization with soft assignment. The concern about limited methodological novelty therefore remains largely outstanding.

Insufficient positioning against prior work: The authors’ responses partially addressed this concern by adding additional experimental comparisons with Q-matrix–related methods and structure-aware KT models , and by arguing that their goal differs from prior Q-matrix optimization. However, clearer conceptual positioning with respect to Q-matrix learning and automatic skill discovery remains unresolved.

Lack of theoretical and empirical justification: The authors’ responses did not introduce new theoretical guarantees beyond empirical monotonic loss reduction, and explicitly acknowledged the heuristic nature of the approach. Claims regarding uncovering cognitive mechanisms were reframed as exploratory but not supported by additional quantitative validation. This concern remains unresolved.

Practical concerns: The authors’ responses addressed computational cost by providing concrete runtime and memory measurements as well as sensitivity analyses. However, limitations regarding unseen questions and the interpretability of the learned groups for educational practice were acknowledged rather than resolved, and thus remain outstanding.

**Reviewer Scores:**

While the discussion may have reduced uncertainty for some reviewers, it is unlikely that it would have led to a meaningful upward shift in scores, particularly for the more critical reviews.

---

### Decision · Program_Chairs · 2026-01-26

Reject